# Effects of L-Methionine and DL-Methionine on Growth Performance, Methionine-Metabolizing Enzyme Activities, Feather Traits, and Intestinal Morphology of Medium-Growing, Yellow-Feathered Chickens between 1 and 30 Days of Age

**DOI:** 10.3390/ani14142135

**Published:** 2024-07-22

**Authors:** Xiajing Lin, Dong Ruan, Zeling Lin, Taidi Xiong, Sheng Zhang, Qiuli Fan, Xiaoli Dong, Yuanfan Deng, Zongyong Jiang, Shouqun Jiang

**Affiliations:** 1Institute of Animal Science, Guangdong Academy of Agricultural Sciences, State Key Laboratory of Swine and Poultry Breeding, Key Laboratory of Animal Nutrition and Feed Science in South China, Ministry of Agriculture and Rural Affairs, Guangdong Provincial Key Laboratory of Animal Breeding and Nutrition, Guangzhou 510640, China; linxiajing728@sohu.com (X.L.); donruan@126.com (D.R.); dering88@126.com (Z.L.); abxtd1997@163.com (T.X.); zhangszin@163.com (S.Z.); fanqiuli_829@163.com (Q.F.); jiangzy@gdaas.cn (Z.J.); 2CJ International Trading Co., Ltd., Shanghai 201107, China; xiaoli.dong@cj.net (X.D.); yuanfan.deng@cj.net (Y.D.)

**Keywords:** methionine, feather trait, relative bioavailability, Met-metabolizing enzymes, broiler chickens

## Abstract

**Simple Summary:**

Methionine (Met), as the first limiting amino acid in a broiler’s diet, plays an important role in protein synthesis, methyl donor, and cell proliferation. Therefore, determining the accurate requirement of Met is of great importance for improving broiler production. However, the optimal L-Met requirement of yellow-feathered broilers remains undetermined at different growth rates. The aim of the trial was to explore the effects of L-Met on growth performance, feather traits, Met-metabolizing enzyme activities, bioavailability, and intestinal morphological characteristics of medium-growing, yellow-feathered broiler chickens compared with DL-Met during the starter phase. Based on the findings, dietary supplementation with L-Met or DL-Met has beneficial effects on the growth performance, feather traits, and intestinal morphology of medium-growing, yellow-feathered broilers aged 1 to 30 d by decreasing the enzyme activities of Met methylation (MAT) and increasing the enzyme activities of the sulfur transfer pathway (CBS), and supplementation with L-Met served a better function compared with DL-Met. The relative efficacy of L-Met to DL-Met was 142.5% for average daily gain (ADG) of yellow-feathered broilers. The appropriate dietary Met levels for medium-growing, yellow-feathered broilers are between 0.36~0.38% (supplementation with DL-Met) or 0.32~0.33% (supplementation with L-Met) when based on ADG and feed-to-gain ratio.

**Abstract:**

This experiment investigated the effects of L-Methionine (L-Met) on growth performance, Met-metabolizing enzyme activity, feather traits, and small intestinal morphological characteristics, and compared these with DL-Methionine (DL-Met) for medium-growing, yellow-feathered broilers during the starter phase. Furthermore, the aim was to provide recommendations for the appropriate dietary Met levels in feed. A total of 1584 1-d broilers were randomly divided into 11 treatment groups with six replicates of 24 birds each: basal diet (CON, Met 0.28%), basal diet + L-Met (0.04%, 0.08%, 0.12%, 0.16%, 0.20%), and basal diet + DL-Met (0.04%, 0.08%, 0.12%, 0.16%, 0.20%). The total trial period was 30 days. Compared with broilers fed the basal diet, those fed 0.04 to 0.20% supplemental Met had higher final body weight (FBW), average daily feed intake (ADFI), average daily gain (ADG), and lower feed-to-gain ratio (F: G) (*p* < 0.05). Compared with DL-Met groups, the L-Met group had higher FBW and ADG (*p* < 0.05). The relative bioavailability (RBV) of L-Met in ADG of 1–30 d was 142.5%. Chicks fed diets supplemented with L-Met had longer fourth primary feather lengths compared to birds fed the control and diets supplemented with DL-Met (*p* < 0.05). Compared to the control, birds supplemented with DL-Met or L-Met had an increased moulting score (*p* ≤ 0.05). Chicks fed diets supplemented with L-Met had lower activities of methionine adenosyl transferase (MAT) compared to those fed the basal diet or supplemented with DL-Met (*p* < 0.05). Chicks supplemented with either DL-Met or L-Met had higher activities of cystathionine β-synthase (CBS) than those fed the basal diet (*p* < 0.05). Compared with the control, chicks fed diets supplemented with either DL-Met or L-Met had an enhanced level of albumin in plasma (*p* < 0.05). There were no obvious differences in the plasma content of uric acid and total protein among the treatments (*p* > 0.05). Chicks fed diets supplemented with either DL-Met or L-Met had higher villus height and V/C in the duodenal than chicks fed the basal diet (*p* < 0.05). The jejunum morphology was not affected by either L-Met or DL-Met supplementation (*p* > 0.05). Therefore, dietary supplementation with DL-Met or L-Met improved the growth performance, feather traits, and intestinal morphological characteristics of medium-growing, yellow-feathered broiler chickens aged 1 to 30 d by decreasing the enzyme activities of Met methylation (MAT) and increasing the enzyme activities of the sulfur transfer pathway (CBS), and supplementation with L-Met showed a better improvement compared with DL-Met. The relative efficacy of L-Met to DL-Met was 142.5% for ADG of yellow-feathered broilers. The appropriate Met levels for medium-growing, yellow-feathered broilers are between 0.36~0.38% (supplementation with DL-Met) or 0.32~0.33% (supplementation with L-Met) when based on ADG and feed-to-gain ratio.

## 1. Introduction

Methionine (Met), as the first limiting amino acid in a broiler’s diet, has multiple biological functions in broiler chickens and plays an important role in protein synthesis [1], methyl donor [2], and cell proliferation [3]. Diets with inadequate or an excess of Met can affect growth performance [4], carcass quality [5], antioxidant capacity [6] (Martinez et al., 2017), and lipid metabolism [7]. A lack of methionine will inhibit the synthesis of proteins in the body, but adequate supplementation can inhibit protein degradation and promote growth [8]. Xue et al. [9] reported that excess DL- and L-Met were toxic for starter Pekin ducks. Therefore, determining the accurate requirement of Met is of great importance for improving broiler production [10]. Currently, DL-methionine (DL-Met) is the most commonly used form of Met in production, which consists of an equal amount of the isomers D-methionine (D-Met) and L-methionine (L-Met) generated through chemical synthesis. D-Met needs to be converted to L-Met by D-Amino acid oxidase (D-AAO) to be utilized [11]. Different dietary Met sources have different Met absorption, and DL-Met may increase apical Met transport in the small intestinal segments, however, DL-2-hydroxy-4-Methylthiobutanoic acid may increase basolateral Met transport in the small intestine and other gastrointestinal tracts [12,13,14,15]. The absorption rate of D-Met and DL-Met in the gut is significantly slower than that of L-Met because D-Met and L-Met compete for many of the same transporters, and these transporters have a much lower affinity for D-Met than L-Met [16]. The beneficial effects of L-Met have been verified in recent research of fast-growing chickens compared with DL-Met [17] and other species [4,18,19]. By contrast, in laying hens, DL-Met showed similar biological efficacy as L-Met for body weight and feather growth [20]. China has numerous native chicken breeds and local consumers prefer the flavour of indigenous meat-type chickens. These studies have shown the advantages of L-Met over DL-Met, although inconsistent observations still exist. Improved yellow-feathered chickens are a primary type of chicken in China, of which 4 billion are consumed annually [21], and consumers prefer indigenous chicken breeds over the rapidly increasing imported broilers. However, the appropriate L-Met requirement of yellow-feathered broilers remains undetermined at different growth rates.

Consequently, the objective of the trial was to investigate the beneficial effects on growth performance, feather quality, Met-metabolizing enzyme activities, bioavailability, and intestinal morphology of medium-growing, yellow-feathered broilers compared with DL-Met during the starter phase.

## 2. Materials and Methods

### 2.1. Birds, Experimental Design, and Diets

This study was approved by the Animal Care Committee of the Institute of Animal Science, Guangdong Academy of Agriculture Science, Guangzhou, China, with the approval number GAASIAS-2022-020. The test site was located in the experimental chicken farm of the Institute of Animal Science, Guangdong Academy of Agricultural Sciences, and the experiment was started on 6 July 2022.

A total of 1584 1-d female *Qingyuan* partridge chickens [22], obtained from Zhongshan Kelang Agricultural Science and Technology Co., Ltd., Zhongshan, China, were randomly divided into 11 treatment groups, and each group had 6 replicates and 24 birds per replicate. Birds in the control group (CON) were fed a basal diet (CON, Met 0.28%), and birds in the other 10 treatment groups received the basal diet with added L-Met (0.04%, 0.08%, 0.12%, 0.16%, 0.20%) or DL-Met (0.04%, 0.08%, 0.12%, 0.16%, 0.20%), respectively (Table 1). Birds were raised on deep litter (wood shavings), with the stocking density of 0.20 m^2^/bird. The house temperature was kept at 32 to 34 °C for the first 3 days and then reduced by 2 °C per week until settled at 28 °C. Diets and water were supplied ad libitum throughout.

Diets based on corn–soybean meal were calculated to meet nutrient requirements for yellow chickens according to the China National Feeding Standard [23], with the exception of added Met. Details of ingredient composition and nutrient contents of the experimental diets for chickens are provided in Table 2. DL-Met and L-Met (≥99% purity) were obtained from CJ (Shanghai, China) Trading Co., Ltd. (Shanghai, China). The additional Met was added to the diet by premix, and dietary Met and cystine were analyzed according to the China National Standard (GB/T 15399-2018) [24]. Other amino acids were analyzed according to the China National Standard (GB/T 18246-2019) [25].

### 2.2. Measurement of Growth Performance

Initial and final body weights were recorded per replicate on d 1 and 30 of age, respectively. Average daily weight gain (ADG) and average daily feed intake (ADFI) were obtained, and feed-to-gain ratio (F:G) was calculated on a replicate basis from 1 to 30 d.

### 2.3. Evaluation of Feather Traits

At the age of 29 d, 30 chickens (5 birds per replicate) were randomly chosen from each treatment group for feather status measurement. Fourth primary feather length measurements were taken with a rule of a minimum scale of 1 mm. The values of abdomen, back, and breast feather measurements were taken following the criteria as shown in Table 3, using the method by Zhang et al. [4].

### 2.4. Sample Collection

At the end of the trial (d 30), 2 birds with close-to-average body weight per replicate were deprived of feed for 12 h, and 5 mL heparinized blood was taken from the brachial vein through an evacuation tube, and plasma was then obtained after centrifuging (3000× *g*, 15 min) at 4 °C. The birds were electrically stunned and subsequently euthanized by exsanguination.

### 2.5. Measurement of Methionine Metabolizing Enzymes and Biochemical Indices in Plasma

The plasma contents of uric acid, total protein, and albumin were assayed with commercial kits (Nanjing Jiancheng Institute of Bioengineering, Nanjing, China), and the plasma activities of betaine-homocysteine methyltransferase (BHMT), glycine n-methyltransferase (GNMT), cystathionine β-synthase (CBS), and methionine adenosyl transferase (MAT) were determined with ELISA kits (Equation biotechnology Co., Ltd., Beijing, China) using a spectrophotometer (Spectra Max M-5, Molecular Devices, San Jose, CA, USA).

### 2.6. Measurement of Intestinal Morphology

Duodenal and jejunal tissue samples of broilers fed with L-Met (0.08%, 0.12%), DL-Met (0.08%, 0.12%), and control diets were collected. Amounts of 1 cm cut from the medial portions of the duodenum and jejunum were washed in physiological saline solution and fixed in 10% buffered formalin. Duodenal and jejunum samples were dehydrated and embedded in paraffin, and then cut into 5 μm sections and routinely stained with hematoxylin-eosin staining (H&E) hematoxylin dye for 5 min and eosin dye for 15 s. Slides were blindly evaluated microscopically (Eclipse Ti-E, Nikon, Japan). Villus height and crypt depth were observed under the scanning–browsing software (CaseViewer 2.2, 3DHISTECH, Budapest, Hungary) and scanning analysis software (Halo v3.0.311.314, Indica Labs, Albuquerque, NM, USA). The ratio of villus height to crypt depth (V/C) was calculated.

### 2.7. Statistical Analysis

Replicate or individual bird was taken as the experimental unit. The effects of dietary Met treatment were analyzed using one-way ANOVA with GLM procedures of SAS (version 8.0). The effects of Met sources (CON vs. DL-Met, CON vs. L-Met, and DL-Met vs. L-Met) and the average of 5 supplemental levels were evaluated using preplanned contrasts [4,17]. Values are presented as means with standard error of mean (SEM). *p* < 0.05 was considered significant. For key performance variables (ADG and F:G), the dietary Met requirement (0, 0.04, 0.08, 0.12, 0.16) of the birds was estimated using quadratic polynomial (QP) and broken-line (BL with plateau) regression models by the NLIN procedure of SAS. QP model: Y = c + bX + aX^2^, where a = quadratic coefficient, b = linear coefficient, c = intercept. The requirement of Met was defined as Met = −b/(2 × a). BL: Broken line; BL with plateau: Y = a + b × Met, Met ≤ c; Y = a + b × c, Met > c, where Y is the dependent variable and X is the dietary Met supplementation level, a is the intercept, b is the slope of line, the value (a + b × c) is the plateau. The Met supplementation level at the break point (c) was considered as the one providing the maximum or minimum of Met supplemental level. The relative bioavailability (RBV) of L-Met and DL-Met was evaluated using the nonlinear exponential regression model analysis [26,27]. Y = a + b × (1 − EXP − (c_1×1_ + c_2×2_)), in which y = variable (ADG and F:G), a = intercept (value for the CON), b = asymptotic response, a + b = common asymptote (maximum growth performance level), c_1_ = slope ratio for DL-Met, c_2_ = slope ratio for L-Met, and X_1_ and X_2_ = dietary supplemental level of DL-Met and L-Met, respectively. The RBV of L-Met and DL-Met were given by the ratio of their c values = [100 × (c_2_/c_1_)] according to Littell et al. [26].

## 3. Results

### 3.1. Growth Performance

As shown in Table 4, the dietary addition of DL-Met and L-Met had higher FBW, ADFI, and ADG, and lower F: G of chicks at 30 d of age (*p* < 0.05), with 0.04% to 0.20% additional Met increasing FBW, ADFI, and ADG, and decreasing F: G (*p* < 0.05) compared with the control group. Chicks’ supplementation with L-Met had higher FBW and ADG than those of chicks’ supplementation with DL-Met. There were no significant differences between the two sources of Met in ADFI and F: G (*p* > 0.05). Higher efficiency of L-Met relative to DL-Met was found in ADG with values of 145.2%, but no appropriate regression could be used in the F:G (Table 5).

As shown in Table 6, the data of ADG and F: G were selected using QP and BL regressions for analysis to obtain the appropriate levels of supplemental DL-Met and L-Met of yellow-feathered chicks from the regressions. The appropriate DL-Met level estimated for ADG using the QP regression is 0.44%, or 0.38% using the BL with the plateau model. The appropriate L-Met level estimated for ADG using the QP regression was 0.43%, or 0.33% using the BL with the plateau model. The appropriate DL-Met levels estimated for F: G using QP and BL with plateau models are 0.46% and 0.36%, respectively. The appropriate L-Met levels estimated for F: G using QP and BL with plateau models are 0.40% and 0.32%, respectively.

### 3.2. Feather Traits

As shown in Table 7, Chicks supplemented with L-Met had longer fourth primary feather lengths than the control birds or those supplemented with DL-Met (*p* < 0.05). Compared to control birds, supplemental DL-Met or L-Met both increased the moulting score (*p* < 0.05), which tended to be higher in chicks supplemented with L-Met rather than DL-Met (*p* = 0.07).

### 3.3. Plasma Biochemical Variables

As presented in Table 8, chicks supplemented with L-Met had lower activities of MAT compared to control birds or those supplemented with DL-Met (*p* < 0.05). Chicks fed a diet supplemented with either DL-Met or L-Met had higher activities of CBS than those fed the basal diet (*p* < 0.05). The activity of GNMT and BHMT was not affected by dietary Met levels and sources (*p* > 0.05). Chicks fed either DL-Met or L-Met had enhanced levels of albumin in their plasma compared to the control birds (*p* > 0.05). There were no differences in the plasma content of uric acid and total protein among the treatments (*p* > 0.05).

### 3.4. Small Intestinal Morphology

As shown in Table 9, Chicks supplemented with either DL-Met or L-Met had higher villus height and V/C in the duodenal than chicks fed the basal diet (*p* < 0.05). The jejunum morphology was not affected by either L-Met or DL-Met supplementation (*p* > 0.05).

## 4. Discussion

Methionine is an essential amino acid for animals and an important substance for protein synthesis [28]. Studies have shown that properly increasing dietary Met levels can significantly improve ADG and F:G of broilers [8,29,30]. Esteve-Garcia and Khan [31] also revealed that broilers fed diets with a continuous increase in the concentration of dietary DL-Met or L-Met had a steady increase in ADFI. In the current study, dietary Met supplementation improved the ADFI, ADG, and F:G of medium-growing, yellow-feathered broilers during the starter stage, and the growth performance of L-Met supplementation groups was significantly higher than that of DL-Met supplementation groups, which suggested that different efficacy may exist between L-Met and D-Met. L-methionine can be directly absorbed and utilized by the body, while D-Met is limited by D-AAO and needs to be transformed into L-Met to be utilized. The current study showed that the efficiency of L-Met relative to DL-Met was 145.2% for ADG, which concurs with some previous studies. For example, Shen et al. [17] reported that the RBV of L-Met relative to D-Met was 140.7% in chickens as calculated by F:G using a slope ratio method. Zhang et al. [4] found that the RBV of L-Met in Cherry Valley ducks relative to DL-Met was 120–140% based on ADG and F:G. Asasi et al. [32] reported that linear regression analysis revealed that DL-Met was 94.97% (for ADG) and 95.63% (for FCR) as efficacious as L-Met. Together, these results suggest that dietary supplementation with Met may improve growth performance of medium-growing, yellow-feathered chickens in the starter stage, and that L-Met has a better bioavailability than DL-Met.

Feathers are mostly keratin, and the properties of keratin depend on the amount of sulfur-containing amino acids, including Met and cystine [33]. The current results demonstrated that dietary supplementation with Met increased the fourth primary feather length and moulting score compared to the control birds, indicating that inadequate Met intake may impair the feather growth of broilers. These findings are consistent with those by Zeng et al. [34], who found that dietary supplementation with Met improved the feather coverage and the fourth primary feather length. It was also reported that diets supplemented with Met improved the total and relative feather coverage and weight of ducks [35]. This may be related to the fact that Met was partly transformed into cysteine, could promote feather development, and improve the sulfur content and keratin composition of feathers. It is noteworthy that L-Met resulted in a better feather trait than DL-Met, which may be attributed to the insufficiency of D-AAO in broiler starter growing phase, leading to a reduced conversion of D-Met to L-Met [11]. Thus, it should be recommended that dietary supplementation with Met be used to improve feather growth of medium-growing, yellow-feathered broilers, and L-Met is more efficacious than DL-Met in this regard.

Methionine is involved in protein synthesis mainly through the metabolic enzymes MAT, BHMT, and CBS. Methionine is catalyzed by MAT to transform into S-adenosyl-methionine (SAM), which participates in the synthesis of polyamines including spermidine and a variety of bioactive substances such as adrenalin, creatine, choline, and carnitine, the reactions called transmethylation [36]. Sadenosyl methionine is metabolized by transmethylation to produce S-adenosyl-L-homocysteine, which is further degraded to produce homocysteine (Hcy). On the one hand, Hcy, catalyzed by BHMT, is provided with active methyl by betaine or folic acid to produce methionine through remethylation, which is the primary way of Met resynthesis [37]. On the other hand, it is catalyzed by CBS and γ-thionease to produce cysteine that participates in the synthesis process of glutathione and other bioactive substances, which is the sulfur transfer pathway of Met metabolism [36]. Methionine deficiency induces tissue to down-regulate CBS protein through an independent mechanism of S-adenosylmethionine to close the thiotransferase pathway, thereby efficiently retaining methionine [38]. In the current study we found that the activity of MAT was decreased and the activity of CBS was increased with Met supplementation. The effects on MAT and CBS were probably due to Met inadequacy, which increases the transmethylation and reduces the trans-sulfur pathway. Similar results were reported by Mato et al. [39], showing that homocysteine can be remethylated into methionine to ensure the normal methylation reaction when methionine is deficient. These results indicate that greater amounts of available Met in vivo are more conducive to Met sulfur transfer, which in turn enhances growth performance and feather traits with dietary Met supplementation. That said, the two forms of Met used also had differences. The total protein level in plasma is a key indicator of protein metabolism [40]. Albumin, the major serum protein, has multiple important physiological functions, including repairing tissue and providing energy [41]. Uric acid concentration in serum is an indicator of protein metabolism and amino acid balance in diets for broilers [42]. When a diet has well-balanced essential amino acids, uric acid concentration in plasma conversely decreases [43]. In the present study, with the increase of Met in the diets, uric acid in plasma increased and then decreased, while albumin presented a linear increase trend, indicating that excessive or inadequate Met levels are not conducive to protein metabolism, but increasing dietary Met levels can increase the content of albumin. The present findings are similar to those of Ho et al. [44], presumably because Met is involved in protein deposition in the body, and the result is consistent with the current results of ADG.

Methionine plays a crucial role in intestinal homeostasis maintenance, and intestinal proliferation and differentiation [45]. Dietary Met supplementation can increase the abundance of jejunal tight junction protein and decrease the activity of caspase-3, a key enzyme of the apoptotic pathway [46]. Teng et al. [47] showed that the appropriate Met level meant better gut integrity and antioxidant status, and L-Met improved growth performance in the starter phase and gut permeability better than DL-Met in the coccidiosis challenge phase. In addition, Met plays a key role in maintaining redox status—its antioxidative function may be another effect on the health of the gastrointestinal tract [27]. In this study, broilers fed diets with DL-Met or L-Met showed improved intestinal morphology in the duodenum, which reflects developing improvement, as manifested through higher villus height and V/C in the duodenum. These results are consistent with those reported by Shen et al. [27], who found that both DL-Met and L-Met supplementation improved villus development in the duodenum of broiler chickens. Similarly, these studies of ducks showed that intestinal morphology was improved by Met supplementation, with no significant difference found between DL-Met and L-Met for this effect [4]. These results indicate that adequate Met is conducive to healthy intestinal development.

## 5. Conclusions

Diets supplemented with L-Met or DL-Met have beneficial effects on the growth performance, feather traits, and intestinal morphological characteristics of medium-growing, yellow chickens aged 1 to 30 d by decreasing the enzyme activities of Met methylation (MAT), and increasing the enzyme activities of the sulfur transfer pathway (CBS); moreover, supplementation with L-Met served a better function compared with DL-Met. The relative efficacy of L-Met to DL-Met was 142.5% for ADG of yellow-feathered broilers. The appropriate dietary Met levels for medium-growing, yellow-feathered broilers are 0.36~0.38% (supplementation with DL-Met) and 0.32~0.33% (supplementation with L-Met) when based on ADG and F:G.

## Figures and Tables

**Table 1 animals-14-02135-t001:** Dietary Met supplemental levels, sources, dietary Met, and Met + Cys contents.

Met Source	Met Supplemental Level, %	Total Dietary Met, % ^1^	Total Dietary Met + Cys, % ^1^
-	0	0.28 (0.28)	0.59 (0.59)
DL-Met	0.04	0.32 (0.30)	0.63 (0.61)
	0.08	0.36 (0.35)	0.67 (0.65)
	0.12	0.40 (0.39)	0.71 (0.70)
	0.16	0.44 (0.44)	0.75 (0.74)
	0.20	0.48 (0.48)	0.79 (0.79)
L-Met	0.04	0.32 (0.32)	0.63 (0.62)
	0.08	0.36 (0.34)	0.67 (0.64)
	0.12	0.40 (0.40)	0.71 (0.70)
	0.16	0.44 (0.42)	0.75 (0.72)
	0.20	0.48 (0.48)	0.79 (0.78)

^1^ The number in parentheses is the analyzed value.

**Table 2 animals-14-02135-t002:** Composition and nutrient levels of the basal diet (g/kg, as fed basis).

Ingredients		Nutrient Levels ^3^	
Corn (CP 7.83%) ^1^	603.9	ME(MJ/kg)	11.92
Soybean meal (CP 46.22%) ^1^	250.05	CP	199.0
Peanut meal (CP 47.01%) ^1^	50.0	Ca	10.0
Pea protein powder (CP 77.78%) ^1^	24.1	P	7.4
Soybean oil	27.4	Non-phytate P	4.8
CaHPO_4_	16.8	Lys	10.5
NaCl	3.0	Met	2.8
Limestone	10.9	Met + Cys	5.9
*L*-Lys·HCl	1.00	Thr	7.2
Unified bran	2.40	Trp	2.2
Premix ^2^	10.0	Arg	14.0
Total	1000.0	Ile	7.7
		Val	9.0

^1^ The crude protein levels of feed were analyzed values. ^2^ Premix provided the following per kilogram of the diet during 1 to 30 d of age: VA 12,000 IU, VD_3_ 600 IU, VE 45 IU, VK_3_ 2.5 mg, VB_1_ 1.8 mg, VB_2_ 9.0 mg, VB_6_ 2.8 mg, VB_12_ 16 mg, 50% choline chloride 2600 mg, niacin 42 mg, pantothenic acid 16 mg, folic acid 1.0 mg, biotin 0.12 mg, Fe 80 mg, Cu 18.8 mg, Mn 60 mg, Zn 80 mg, I 0.7 mg, Se 0.15 mg. ^3^ ME and Non-phytate P was a calculated value based on the data presented in the Chinese feed database (Chinese feed database. 2021) while the others were measured values.

**Table 3 animals-14-02135-t003:** Scoring criteria of feather traits ^1^.

Molting Degree	Feather Class	Score
The back, breast, and abdomen have not yet moulted.	0−	1
	0	2
	0+	3
A few feathers on the back, breast, and abdomen began to molt.	1−	4
	1	5
	1+	6
The back, breast, and abdomen are moulting, and there are more down feathers left.	2−	7
	2	8
	2+	9
The back, breast, and abdomen are moulting more, and there are fewer down feathers left.	3−	10
	3	11
	3+	12

^1^ According to the reference method by Zhang et al. [4].

**Table 4 animals-14-02135-t004:** Effects of Met sources and supplemental levels on growth performance of yellow-feathered broilers from 1 to 30 days of age ^1^.

Variable		IBW (g)	30 d FBW (g)	ADFI (g/d)	ADG (g/d)	F:G
Met source	Supplemental Dietary Met level					
CON	0.00	38.79	486.08	31.81	15.97	2.00
DL-Met	0.04	38.81	563.85	34.75	18.75	1.87
	0.08	38.81	570.09	34.40	18.97	1.82
	0.12	38.78	616.85	36.99	20.73	1.82
	0.16	38.83	579.86	33.87	19.30	1.76
	0.20	38.82	613.20	34.99	20.51	1.71
L-Met	0.04	38.82	587.50	35.06	19.57	1.79
	0.08	38.79	587.42	33.92	19.61	1.74
	0.12	38.80	597.45	34.60	19.95	1.74
	0.16	38.85	626.03	36.41	20.97	1.77
	0.20	38.84	618.53	36.03	20.70	1.76
SEM		0.03	9.31	0.63	0.33	0.03
DL-Met ^2^		38.81	589.23	35.00	19.65	1.80
L-Met ^3^		38.82	603.31	35.20	20.16	1.76
*p*-values						
CON vs. DL-Met		0.54	<0.01	<0.01	<0.01	<0.01
CON vs. L-Met		0.39	<0.01	<0.01	<0.01	<0.01
DL-Met vs. L-Met		0.67	0.02	0.61	0.02	0.07

^1^ Data are means of six replicates per treatment (24 birds per replicate). IBW = Initial body weight; FBW = Final body weight; ADFI = Average daily feed intake; ADG = average daily gain; F:G = Feed-to-gain ratio. ^2^ DL-Met = average of five supplemental levels of DL-Met. ^3^ L-Met = average of five supplemental levels of L-Met.

**Table 5 animals-14-02135-t005:** The RBV of L-Met in comparison to DL-Met of ADG in yellow-feathered broilers from 1 to 30 days of age.

Variable	Regression Equation ^1^	R^2^	RBV	95% Confidence Intervals
ADG	Y = 16.03 + 4.47 × [1 − Exp − (19.09 X_1_ + 27.72 X_2_)]	0.93	145.2%	0.12~0.38

ADG = average daily gain; RBV = relative bioavailability. ^1^ Y = a + b × (1 − EXP − (c_1×1_ + c_2×2_)), in which y = variable (ADG), a = intercept (value for the CON), b = asymptotic response, a + b = common asymptote (maximum growth performance level), c_1_ = slope ratio for DL-Met, c_2_ = slope ratio for L-Met, and X_1_ and X_2_ = dietary supplemental level of DL-Met and L-Met, respectively. The RBV of L-Met and DL-Met were given by the ratio of their c values = [100 × (c_2_/c_1_)].

**Table 6 animals-14-02135-t006:** Optimum Met supplementation of yellow-feathered broilers from 1 to 30 days of age.

Variables	Met Source	Model	Regression Equation ^1^	R^2^	*p* Value	Met Supplemental Level, %	Met Requirement ^4^,%	Met Requirement, mg/d
ADG	DL-Met	QP ^2^	Y = −165.91 × X^2^ + 52.76 × X + 16.27	0.76	<0.01	0.16	0.44	152
		Two-slope BL ^3^	Y = 16.40 + 37.50 × X, X < 0.10, Y = 20.19, X ≥ 0.10	0.71	<0.01	0.10	0.38	131
	L-Met	QP	Y = −167.25 × X^2^ + 54.46 × X + 16.40	0.76	<0.01	0.15	0.43	149
		BL with plateau	Y = 15.97 + 83.87 × X, X < 0.05, Y = 20.31, X ≥ 0.05	0.76	<0.01	0.05	0.33	112
F:G	DL-Met	QP	Y = 7.54 × X^2^ − 2.91 × X + 1.99	0.87	<0.01	0.18	0.46	168
		BL with plateau	Y = 2.00 − 3.21 × X, X < 0.08, Y = 1.76, X ≥ 0.08	0.77	<0.01	0.08	0.36	124
	L-Met	QP	Y = 14.68X^2^ − 3.85X + 1.97	0.77	<0.01	0.12	0.40	138
		BL with plateau	Y = 2.00 − 1.00 × X, X < 0.04, Y = 1.76, X ≥ 0.04	0.84	<0.01	0.04	0.32	110

ADG = average daily gain; F: G = Feed-to-gain ratio. ^1^ Regression equations obtained using the analyzed Met (DL-Met or L-Met) in the trial diets (0, 0.04, 0.08, 0.12, 0.16 and 0.20%). ^2^ QP: Quadratic polynomial; QP model: Y = c + bX + aX^2^, where a = quadratic coefficient, b = linear coefficient, c = intercept. Y is the dependent variable and X is the dietary Met supplemental levels. The maximum or minimum of Met supplemental level was obtained by-b/(2 × a). ^3^ BL: Broken line; BL with plateau: Y = a + b × Met, Met ≤ c; Y = a + b × c, Met > c, where Y is the dependent variable and X are the dietary Met supplemental levels, a is the intercept, b is the slope of line, the value (a + b × c) is the plateau. The Met supplemental level at the break point (c) was considered as the one providing the maximum or minimum of Met supplemental level. The reference values were considered as 0.32%, 0.33%, and 0.36%. ^4^ Met requirement, % = Met supplemental level + basal diet level (0.28).

**Table 7 animals-14-02135-t007:** Effects of Met source and supplemental levels on feather development of yellow-feathered broilers at 30 days of age ^1.^

Variable		Length of Fourth Primary Feather (cm)	Moulting Score
Met source	Met supplemental level		
CON	0.00	6.34	6.64
DL-Met	0.04	6.85	7.73
	0.08	6.65	7.49
	0.12	6.81	7.52
	0.16	6.85	7.67
	0.20	6.87	7.71
L-Met	0.04	7.03	7.75
	0.08	7.79	8.26
	0.12	6.77	7.52
	0.16	7.00	7.58
	0.20	8.14	10.86
SEM		0.08	0.14
DL-Met ^2^		6.83	7.66
L-Met ^3^		7.35	8.40
*p*-values			
CON vs. DL-Met		0.05	0.02
CON vs. L-Met		<0.01	<0.01
DL-Met vs. L-Met		<0.01	0.07

^1^ Data are means of six replicates per treatment (24 birds per replicate). ^2^ DL-Met = average of five supplemental levels of DL-Met. ^3^ L-Met = average of five supplemental levels of L-Met.

**Table 8 animals-14-02135-t008:** Effects of Met sources and levels on Met metabolism-related enzyme activities and biochemical indices of plasma in 30-day broilers ^1.^

Variable		MAT (pg/mL)	GNMT (ng/L)	BHMT (ng/L)	CBS (pg/mL)	Uric Acid (μmol/L)	Total Protein (μg/mL)	Albumin (μg/mL)
Met source	Met supplemental level							
CON	0.00	16.60	10.44	22.54	141.79	247.09	29.09	8.49
DL-Met	0.04	15.49	10.39	22.11	154.93	251.55	28.71	10.71
	0.08	15.63	11.06	21.93	165.08	275.51	31.78	11.22
	0.12	14.99	10.86	21.75	158.50	310.00	28.00	11.76
	0.16	16.93	10.35	21.36	155.04	301.12	32.49	12.37
	0.20	14.89	10.21	21.18	139.87	280.05	26.39	13.61
L-Met	0.04	16.36	10.27	18.68	135.15	300.08	34.53	11.93
	0.08	14.38	9.97	28.42	169.46	311.17	25.93	12.40
	0.12	14.31	11.02	24.94	161.14	243.08	27.17	13.06
	0.16	13.38	10.06	25.33	158.88	223.46	28.43	13.75
	0.20	13.77	11.45	24.03	170.28	226.99	30.05	13.54
SEM		0.694	0.43	1.14	4.08	23.57	1.28	0.72
DL-Met ^2^		15.61	10.59	22.53	154.29	283.77	29.47	12.19
L-Met ^3^		14.42	10.48	23.96	157.70	258.88	29.22	12.68
*p*-values								
CON vs. DL-Met		0.14	0.75	0.45	<0.01	0.20	0.78	<0.01
CON vs. L-Met		<0.01	0.79	0.13	<0.01	0.68	0.69	<0.01
DL-Met vs. L-Met		<0.01	0.94	0.21	0.06	0.11	0.74	0.32

^1^ Data are means of six replicates per treatment (two birds per replicate). MAT = methionine adenosyltransferase; GNMT = glycine n-methyltransferase; BHMT = betaine-homocysteine methyltransferase; CBS = cystathionine β-synthase; ^2^ DL-Met = average of five supplemental levels of DL-Met. ^3^ L-Met = average of five supplemental levels of L-Met.

**Table 9 animals-14-02135-t009:** Effects of Met source and supplemental levels on intestinal morphology of broilers ^1^.

Variable		Duodenal Villus Height (μm)	Duodenal Crypt Depth (μm)	Duodenal V/C	Jejunum Villus Height (μm)	Jejunum Crypt Depth (μm)	Jejunum V/C
Met source	Met supplemental level						
CON	0.00	1175.26	365.40	3.22	947.84	333.25	3.03
DL-Met	0.08	1344.31	365.75	3.76	892.48	394.28	2.30
	0.20	1413.39	319.66	4.61	959.44	341.21	2.87
L-Met	0.08	1332.96	330.38	4.18	879.90	387.09	2.63
	0.20	1339.45	317.98	4.41	1061.65	304.65	3.59
SEM		24.05	10.51	0.17	66.34	28.62	0.34
DL-Met ^2^		1378.85	342.71	4.19	928.20	365.98	2.60
L-Met ^3^		1335.56	325.42	4.27	957.79	351.76	3.04
*p*-values							
CON vs. DL-Met		<0.01	0.40	0.02	0.81	0.34	0.30
CON vs. L-Met		<0.01	0.17	0.02	0.90	0.59	0.97
DL-Met vs. L-Met		0.35	0.49	0.81	0.69	0.66	0.25

^1^ Data are means of six replicates per treatment (two birds per replicate). V/C = villus height/crypt depth. ^2^ DL-Met = average of two supplemental levels of DL-Met. ^3^ L-Met = average of two supplement levels of L-Met.

## Data Availability

All datasets collected and analyzed during the current study are available from the corresponding author by request, the availability of the data is restricted to investigators based at academic institutions.

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
