# Peer review of "Effects of L-Methionine and DL-Methionine on Growth Performance, Methionine-Metabolizing Enzyme Activities, Feather Traits, and Intestinal Morphology of Medium-Growing, Yellow-Feathered Chickens between 1 and 30 Days of Age"

_animals, 2024, doi:10.3390/ani14142135_

Round 1

Reviewer 1 Report

Comments and Suggestions for Authors

Dear Author, 

The paper addresses a topical issue of the efficacy of metionin in poultry nutrition, in this case - the use of L-Methionine and DL-Methionine in medium-growing yellow- 4 feathered chickens diets. The topic of the paper is very interesting, but the manuscript needs minor revision before it can be accepted for publication. Please see my specific comments attached which might help you to improve the manuscript scientifically: 

- Line 79- Were the chickens used in the experiment sexed?

- Line 93 - Were different types of feed used for particular rearing periods? (e.g. starter)?

- Line 94 - What coccidiostat was used?

- Line 133 - You did not provide staining times for the preparations (e.g. hematoxylin: 5 minutes). 

- Line 228 - You should add reference values. 

Author Response

Comments 1: Line 79- Were the chickens used in the experiment sexed?

Response 1: Thank you for pointing this out. The female chickens were used in the experiment. We have added it in the revised manuscript (Line 116). “A total of 1,584 1-d female”

Comments 2: Line 93 - Were different types of feed used for particular rearing periods? (e.g. starter)?

Response 2: Yes, the nutrient level of the feed varies at each stage (starter, grower, finisher), and the ingredients of the basal diet in each group during the starter period was the same, only the content of Met was not the same.

Comments 3: Line 94 - What coccidiostat was used?

Response 3: The dinitrotomide coccidiostat was used in the experiment.

Comments 4: You did not provide staining times for the preparations (e.g. hematoxylin: 5 minutes).

Response 4: Thank you for pointing this out. We have added“hematoxylin dye for 5 minutes, eosin dye for 15 seconds”in the revised manuscript. (Line 174-175).

Comments 5: Line 228 - You should add reference values.

Response 5: Thank you for pointing this out. We have added reference values in the revised manuscript. (Line 246) “The reference values were considered as 0.32%, 0.33%, and 0.36%.”

4. Response to Comments on the Quality of English Language

Point 1:Extensive editing of English language required

Response 1: Thanks for your suggestion. We have modified it in the revised manuscript.   (in highlighting)

Reviewer 2 Report

Comments and Suggestions for Authors

The novelty and relevance of the research is low. In many doi publications: 10.1016/j.ps.2023.103143, doi: 10.3390/ani12151928, doi: 10.1093/jas/skaa315, doi: 10.3382/ps/pex380, doi: 10.1016/j.psj.2023.102586, doi: 10.1080/00071668.2021.1884653 this problem has been solved. These links are missing from the text of the article.

The abstract and conclusions should be improved. It is necessary to include specific results on the studied indicators.

Author Response

Comments 1: The novelty and relevance of the research is low. In many doi publications: 10.1016/j.ps.2023.103143, doi: 10.3390/ani12151928, doi: 10.1093/jas/skaa315, doi: 10.3382/ps/pex380, doi: 10.1016/j.psj.2023.102586, doi: 10.1080/00071668.2021.1884653 this problem has been solved. These links are missing from the text of the article.

Response 1: Thank you for pointing this out. We have added the references you mentioned in the revised manuscript. (doi: 10.3382/ps/pex380 Line 80; doi: 10.1093/jas/skaa315, doi: 10.1080/00071668.2021.1884653 Line 86-90; doi: 10.3390/ani12151928 Line 96-97; doi: 10.1016/j.ps.2023.103143 Line 304-306; doi: 10.1016/j.psj.2023.102586 Line 365-367.)

Comments 2: The abstract and conclusions should be improved. It is necessary to include specific results on the studied indicators.

Response 2: Thank you for pointing this out. We have modified it in the revised manuscript. (Line 55-58; 380) .

Reviewer 3 Report

Comments and Suggestions for Authors

1. The duplication rate of the manuscript is 40 percent, which is much higher than 20 percent.

2. Manuscripts are not prepared in accordance with journal requirements, e.g., abstracts are larger than 200 words and needs to be further shortened.

3. Line 30: MAT and CBC needs to be written with the full name on the first occurrence.

4. Whats mean “medium-growing yellow-feathered broilers”?

5. Results of the description of intestinal morphology are missing from the Abstract section.

6. Results were supplemented with histopathologic findings of intestinal morphology.

7. The introduction section of the paper suggests the importance of additional amino acid supplementation for broiler production and what role it has. What are the characteristics of MET compared to other amino acids and what is its potential role for broiler production.

8. Focusing on broiler feather characteristics, how does this relate to studied broiler growth performance, gut morphology, etc.?

9. The results in Table 5 show missing.

10. The paper is a study of the effects of L-Methionine and DL-Methionine on broiler chickens? Or is it a study of the effects of L-Methionine on broilers compared to DL-MET? Please express it clearly.

Comments on the Quality of English Language

Extensive editing of English language required

Author Response

Comments 1: The duplication rate of the manuscript is 40 percent, which is much higher than 20 percent.

Response 1: Thank you for pointing this out. We have modified it and lowed the duplication rate to 20% in the revised manuscript.

Comments 2: Manuscripts are not prepared in accordance with journal requirements, e.g., abstracts are larger than 200 words and needs to be further shortened.

Response 2: We have shortened the abstract in the revised manuscript (Line 15-32).

Comments 3: Line 30: MAT and CBC needs to be written with the full name on the first occurrence.

Response 3: Thank you for pointing this out. We have modified it in the revised manuscript. (Line 50; 52)

Comments 4: What’s mean “medium-growing yellow-feathered broilers”?

Response 4: According to the growth rate, yellow-feathered broilers are divided into fast-growing type, medium-growing type and slow-growing type.

Comments 5,6: Results of the description of intestinal morphology are missing from the Abstract section. 6.Results were supplemented with histopathologic findings of intestinal morphology.

Response 5,6: Thank you for pointing this out. We have added the description of intestinal morphology in the revised manuscript. (Line 55-58)

Comments 7: The introduction section of the paper suggests the importance of additional amino acid supplementation for broiler production and what role it has. What are the characteristics of MET compared to other amino acids and what is its potential role for broiler production.

Response 7: Methionine is the first limiting amino acid and has multiple biological functions in broiler chickens, it can affect growth performance, carcass quality, antioxidant capacity and lipid metabolism of broilers. (Line 75-77)

Comments 8: Focusing on broiler feather characteristics, how does this relate to studied broiler growth performance, gut morphology, etc.?

Response 8: Keratin is the main component of poultry feathers, mainly composed of sulfur amino acids. Methionine is the most important amino acid for keratin synthesis, and broilers will consume a lot of energy and protein during molting, which may affect the growth performance of broilers.

Comments 9: The results in Table 5 show missing.

Response 9: The results in Table 5 were shown in Line 227-235.

Comments 10: The paper is a study of the effects of L-Methionine and DL-Methionine on broiler chickens? Or is it a study of the effects of L-Methionine on broilers compared to DL-MET? Please express it clearly.

Response 10: The effects and differences of two different methionines on broilers were studied.

4. Response to Comments on the Quality of English Language

Point 1: Extensive editing of English language required

Response 1: Thanks for your suggestion. We have modified it in the revised manuscript.   (in highlighting)

Reviewer 4 Report

Comments and Suggestions for Authors

L-Methionine and DL-Methionine are two commonly used amino acids in poultry nutrition to improve growth performance and feather quality in birds. Methionine is an essential amino acid that cannot be synthesized by the body and must be obtained through diet. The current manuscript aims to investigate the effects of L-Methionine (L-Met) on growth performance, Met metabolizing enzyme activity, feather traits, and small intestinal morphology for medium-growing yellow-feathered broilers during the starter phase. It is an interesting article that can add to the field, but it needs to be improved and presented better, considering the following comments and suggestions:

- Decrease the similarity percentage (Percent match= 40%!).

- Where is the “Simple summary” section?

- “11 treatment groups”: You employed a large number of treatments rather than focusing on a smaller number to make analysis and follow-up easier.

- Why did you choose only the starter phase for your study?

- L.25: “L-Met group”: Which level exactly?

- L.26: “relative bioavailability”: How did you define this term?

- L.27-28: What is the benefit behind this result?

- L.29: Remove “both”.

- L.30: “MAT”: As you mentioned this abbreviation for the first time in the text, you should define it as “Met methylation”.

- L.32: “CBS”: Same previous comment.

- L.39: “Therelative efficacy”: Adjust the missed-space and kindly inform how you calculated this percentage?

- L.40: I guess you mean “The appropriate”. Correct it.

- L.41: “are between”. Correct.

- L.42: “F:G”: Not clear!

- Keywords: I suggest changing “Medium- growing yellow-feathered chickens” to “broiler chickens”.

- Introduction: Too short and not sufficient! Expand and update it.

- Materials and Methods: Add the study period and location.

- L.78: “Qingyuan partridge chickens”: Add a reference to describe this strain.

- L.83: “with the stocking density of 0.20 m2/bird”: Until when?

- L.84: Replace “room” with “house”.

- L.87: I didn’t find this reference below!

- L.98-99: Carefully revise it.

- L.57-58: “Presence of DL-Met in diet can increase absorption of L-Met in the small intestinal segments and ..”: Expand explaining the scientific reason.

- L.61: “is particularly obvious under heat stress”: Why? Clarify. Also, try to update the reference.

- Statistical Analysis: Revise & Simplify it.

- L.306: “Perry et al., 1987”: Too old!

- L.314-315: Refer to more recent related references.

- Conclusion L.335 and 336: Correct to “0.36-0.38%” and “0.32-0.33%”.

- References: You should follow the journal format.

Comments on the Quality of English Language

Acceptable.

Author Response

Comments 1: Decrease the similarity percentage (Percent match= 40%!).

Response 1: Thank you for pointing this out. We have modified it and lowed the duplication rate to 20% in the revised manuscript.

Comments 2: Where is the “Simple summary” section?

Thank you for pointing this out. We have added it in the revised manuscript (Line 15-32).

Comments 3: “11 treatment groups”: You employed a large number of treatments rather than focusing on a smaller number to make analysis and follow-up easier.

Response 3: We adopted CON+(2 Met resources * 5 Met levels) design, not just only large number of treatments. We classified the comparison as CON vs DL-Met, CON vs L-Met, and DL-Met vs L-Met , so it is easier for readers to understand and get important information.

Comments 4: Why did you choose only the starter phase for your study?

Response 4: The requirement of methionine during each stage needs to design respectively. We have finished three experiments of each stage. Generally, the response of nutrient deficiency is obvious and happened at starter phase. So we have choosed the starter phase for the study.

Comments 5: L.25: “L-Met group”: Which level exactly?

Response 5: We are very sorry, we just classified the comparison as CON vs DL-Met, CON vs L-Met, and DL-Met vs L-Met, there were no specific levels for comparison.

Comments 6: L.26: “relative bioavailability”: How did you define this term?

Response 6: Comparison of biological activity between nutrient sources is usually expressed as relative bioavailability (RBV).

Comments 7: L.27-28: What is the benefit behind this result?

Response 7: Feathers are mostly keratin, and the properties of keratin depend on the amount of sulfur-containing amino acids, including Met and cystine, at the same time, moulting requires energy. These results indicated that the addition of Met could promote the synthesis of keratin and improve the growth performance, while the effect of L-Met was better than that of DL-Met.

Comments 8: L.29: Remove “both”.

Response 8: Thank you for pointing this out. We have deleted it in the revised manuscript. (Line 48).

Comments 9: L.30: “MAT”: As you mentioned this abbreviation for the first time in the text, you should define it as “Met methylation”.

Response 9: Thank you for pointing this out. We have modified it in the revised manuscript. (Line 49-50).

Comments 10: L.32: “CBS”: Same previous comment.

Response 10: Thank you for pointing this out. We have modified it in the revised manuscript. (Line 52)

Comments 11: L.39: “The relative efficacy”: Adjust the missed-space and kindly inform how you calculated this percentage?

Response 11: Thank you for pointing this out. We have added the space in the revised manuscript (Line 63-64), the calculation is in Materials and Methods (Line 197-202).

Comments 12: L.40: I guess you mean “The appropriate”. Correct it.

Response 12: Thank you for pointing this out. We have modified it in the revised manuscript. (Line 65).

Comments 13: L.41: “are between”. Correct.“are between”。

Response 13: Thank you for pointing this out. We have modified it in the revised manuscript. (Line 65).

Comments 14: L.42: “F:G”: Not clear!“F:G”

Response 14: Thank you for pointing this out. We have modified it in the revised manuscript. (Line 67).

Comments 14: Keywords: I suggest changing “Medium- growing yellow-feathered chickens” to “broiler chickens”.

Response 14: Thanks for your suggestion. We have modified it in the revised manuscript. (Line 68-69).

Comments 15: Introduction: Too short and not sufficient! Expand and update it.

Response 15: Thanks for your suggestion. We have expanded the introduction in the revised manuscript (Line 77-80; 86-89; 91-97).

Comments 16: Materials and Methods: Add the study period and location.

Response 16: Thanks for your suggestion. We have added this “The test site was located in the experimental chicken farm of the Institute of Animal Science, Guangdong Academy of Agricultural Sciences, and the experiment was lunched on July 6th, 2022” in the revised manuscript (Line 113-115).

Comments 17: L.78: Qingyuan partridge chickens: Add a reference to describe this strain.

Response 17: Thanks for your suggestion. We have added the reference in the revised manuscript (Line 116-118, Line 490-492 ).

Comments 18: L.83: with the stocking density of 0.20 m2/bird: Until when?

Response 18: The stocking density of 0.20 m2/bird was the entire period of experiment.

Comments 19: L.84: Replace “room” with “house”

Response 19: Thank you for pointing this out. We have modified it in the revised manuscript. (Line 124).

Comments 20: L.87: I didn’t find this reference below!

Response 20: Thank you for pointing this out. We have added the reference in the revised manuscript. (Line 463-464).

Comments 21: L.98-99: Carefully revise it.

Response 21: Thank you for pointing this out. We have checked the levels of vitamins and minerals and modified it in the revised manuscript. (Line 136-140)

Comments 22: L.57-58: Presence of DL-Met in diet can increase absorption of L-Met in the small intestinal segments and ..: Expand explaining the scientific reason.

Response 22: Thanks for your suggestion. We have modified it in the revised manuscript. (Line 86-89).

Comments 23: L.61: is particularly obvious under heat stress: Why? Clarify. Also, try to update the reference.

Response 23: Thanks for your suggestion. We have explained and renewed the references in the revised manuscript. (Line 91-95).

Comments 24: Statistical Analysis: Revise & Simplify it.

Response 24: We are very sorry, there were many mathematic models used in the study, which needed a complete description for the methods, and can not be simplified.

Comments 25: L.306: Perry et al., 1987: Too old!Perry et al.

Response 25: Thanks for your suggestion. We have replaced the reference ”Donsbough et al., 2010” in the revised manuscript. (Line 351-352)

Comments 26: L.314-315: Refer to more recent related references.

Response 26: Thanks for your suggestion. We have replaced the reference in the revised manuscript. (Line 349-351)

Comments 27: Conclusion L.335 and 336: Correct to 0.36-0.38% and 0.32-0.33%.

Response 27: Thank you for pointing this out. We have modified it in the revised manuscript. (Line 386)

Comments 28: References: You should follow the journal format.

Response 28: Thank you for pointing this out. We have modified it in the revised manuscript. (Line 410-529)

4. Response to Comments on the Quality of English Language

Point 1: Acceptable

Response 1: Thank you very much. 

Round 2

Reviewer 2 Report

Comments and Suggestions for Authors

Correction of the article based on these comments is accepted.

Author Response

Comments 1: Correction of the article based on these comments is accepted.

Response 1: Thank you!

Reviewer 3 Report

Comments and Suggestions for Authors

Ensure that the references are meticulously verified for uniformity in their formatting, and carefully review the citation style to align with the journal's guidelines.

Comments on the Quality of English Language

English quality is good enough to receive publication

Author Response

Comments 1: Ensure that the references are meticulously verified for uniformity in their formatting, and carefully review the citation style to align with the journal's guidelines.

Response 1: Thanks for your suggestion. We have modified references in the revised manuscript.

Reviewer 4 Report

Comments and Suggestions for Authors

Accept.

Comments on the Quality of English Language

Acceptable.

Author Response

Comments 1: Accept.

Response 1: Thanks!